# A Novel Strategy for Constructing an Integrated Linkage Map in an F_1_ Hybrid Population of *Populus deltoides* and *Populus simonii*

**DOI:** 10.3390/genes13101731

**Published:** 2022-09-26

**Authors:** Zhiting Li, Jinpeng Zhang, Zhiliang Pan, Shengjun Bai, Chunfa Tong

**Affiliations:** Co-Innovation Center for Sustainable Forestry in Southern China, College of Forestry, Nanjing Forestry University, Nanjing 210037, China

**Keywords:** genetic linkage map, next-generation sequencing, *Populus*, F_1_ hybrid population, outbred species

## Abstract

The genetic linkage maps of the traditional F_2_ population in inbred lines were estimated from the frequency of recombination events in both parents, providing full genetic information for genetic and genomic studies. However, in outbred forest trees, it is almost impossible to generate the F_2_ population because of their high heterozygosity and long generation times. We proposed a novel strategy to construct an integrated genetic linkage map that contained both parental recombination information, with restriction-site-associated DNA sequencing (RADSeq) data in an F1 hybrid population of *Populus deltoides* and *Populus simonii*. We selected a large number of specific RAD tags to construct the linkage map, each of which contained two SNPs, one heterozygous only in the female parent and the other heterozygous only in the male. Consequently, the integrated map contained a total of 1154 RAD tags and 19 linkage groups, with a total length of 5255.49 cM and an average genetic distance of 4.63 cM. Meanwhile, the two parent-specific linkage maps were also constructed with SNPs that were heterozygous in one parent and homozygous in the other. We found that the integrated linkage map was more consensus with the genomic sequences of *P. simonii* and *P. deltoides*. Additionally, the likelihood of the marker order in each linkage group of the integrated map was greater than that in both parental maps. The integrated linkage map was more accurate than the parent-specific linkage maps constructed in the same F_1_ hybrid population, providing a powerful genetic resource for identifying the quantitative trait loci (QTLs) with dominant effects, assembling genomic sequences, and performing comparative genomics in related *Populus* species. More importantly, this novel strategy can be used in other outbred species to build an integrated linkage map.

## 1. Introduction

Genetic linkage maps display the linear orders of groups of DNA markers with genetic distances between them, playing a crucial role in identifying the quantitative trait loci (QTLs), assembling genomic sequences, and performing comparative genomics [1,2,3,4]. For experimental plants and animals in which inbred lines are available, the simplest populations for constructing linkage maps are the backcross (BC) and F_2_ intercross between two inbred lines [5]. Mather (1938) showed that the F_2_ populations provided more recombination information than the backcross because two, rather than one, recombinant gametes are possibly sampled per individual [6]. Thus, when the sample sizes are the same, the linkage map constructed with an F_2_ population is more accurate than that with a backcross, in the aspects of the genetic distances between the adjacent markers and marker orders in each linkage group. Without any doubt, accurate linkage maps are more powerful for locating QTLs and anchoring chromosome-scale sequences [1,7]. In addition, linkage maps that contain both parental recombinant information allow for the effective identification of QTLs with both additive and dominant effects [8,9].

However, for outbred species, especially in forest trees, it is almost impossible to derive the F_2_ populations as in the inbred lines, due to their long generation times and high heterozygosity [10,11]. Therefore, an F_1_ hybrid population was usually established by crossing two diverged individuals as a mapping population, and the so-called “pseudo-testcross” strategy was applied for linkage mapping, leading to two independent parental linkage maps [1,12,13,14,15]. Each of the two linkage maps contains recombinant information only from one parent. Although efforts have been made to construct an integrated linkage map in an outbred F_1_ hybrid population [16,17,18,19], successful cases similar to the F_2_ maps in inbred lines have rarely been reported to date. The main reason is that the overwhelming majority of molecular markers segregate in the ratio of 1:1 in the F_1_ progeny, at which one parent is heterozygous and the other homozygous, but there were no sufficient markers segregating as bridges in both parents for generating a complete linkage map [1,20]. Meanwhile, many studies have been performed by combining multiple linkage maps into an integrated map for saturation purposes or for comparative genomics [10,21,22,23]. However, these maps could not have the full characteristics of the F_2_ linkage maps in inbred lines.

With the advances in next-generation sequencing technologies, a large number of single nucleotide polymorphisms (SNPs) can be obtained in a fast and cheap way across a hybrid population for genetic mapping. These SNPs provide opportunities for mining special markers to perform various genetic studies. In the current study, we proposed a novel strategy to construct an integrated linkage map that contained both parental recombinant information between the adjacent markers in an F_1_ hybrid population of *P. deltoides* and *P. simonii*. In our previous studies [20,24], a large number of the hybrids were sequenced with the restriction-site-associated DNA sequencing (RAD-seq) technology. The SNP genotypes of the two parents and some hybrids were identified by aligning their paired-end (PE) reads to the reference genome of *P. simonii* [25]. We chose a large number of specific RAD tags to construct the integrated map, each of which contained two SNPs, one heterozygous only in the female parent and the other heterozygous only in the male (Figure 1A), to construct an integrated linkage map of *P. deltoides* and *P. simonii*. Such a linkage map differed from those constructed with the “pseudo-testcross” strategy in forest trees, providing full genetic information for locating quantitative trait loci (QTLs), assembling genomic sequences, and performing comparative genomics. Furthermore, the proposed strategy could be applied to other outbred species for constructing an integrated linkage map with an F_1_ hybrid population.

## 2. Materials and Methods

### 2.1. Plant Material and Sequencing Data

We randomly selected 257 trees in an F_1_ hybrid population for constructing the parental integrated linkage map. The F_1_ population was derived by crossing a female *P. deltoides* and a male *P. simonii* in the springs of 2009 and 2011, as described in our previous studies [20,24]. The selected individual trees and their two parents were sequenced with RAD-seq technology in 2013 and 2016. The read data for each tree had a genomic coverage of >2x, and its accession number is listed in Appendix A. All the read data are available in the NCBI SRA database (http://www.ncbi.nlm.nih.gov/Traces/sra, accessed on 1 May 2022).

### 2.2. SNP Genotyping

First, the PE read data of each tree was filtered to produce high-quality (HQ) data with the NGS QC toolkit (v2.3.3) [26]. Next, we aligned the HQ reads of each individual to the reference sequence of *P. simonii* [25] to call its SNP genotypes using the software packages of BWA (v0.7.17) [27], SAMtools (v1.9), and BCFtools (v1.9) [28]. The whole SNP calling procedures can be described in detail as follows: (1) The HQ reads of each tree were mapped to the reference genome to generate a sequence alignment/map (SAM) file with the BWA *mem* command; (2) each SAM file was converted into a BAM file and then was sorted and indexed with SAMtools; (3) each sorted BAM file was converted into a BCF file using BCFtools with its parameters set as follows: “*bcftools mpileup -Obuzv -a AD,INFO/AD –f*”; (4) for each parent, a VCF file was obtained from its BCF file with the command as “*bcftools call-m -v -f gq*”; (5) the SNPs of each parent were extracted from their VCF files and merged into a site list file, in which the allele read coverage depth (DP) of each genotype at each SNP site was ≥3, and the corresponding genotyping quality (GQ) was >30; (6) for each progeny, a VCF file was derived from its BCF file according to the site list file obtained above, with the command as “*bcftools call-m -v -f gq*; (7) the SNP genotypes for all individuals were extracted from their VCF files such that the allele DP ≥ 3 and GQ > 30, finally leading to a SNP dataset.

A chi-square test was performed for each SNP to check whether it follows the Mendelian segregation ratio. If an SNP seriously deviated from the Mendelian ratio (*p* < 0.01) or had ≥20% missing genotypes, it was removed from the dataset.

### 2.3. Integrated Linkage Map Construction

As a novel strategy, we selected the RAD tags (<1000 bp) that contain two SNPs to construct an integrated linkage map of *P. deltoides* and *P. simonii*. One of the SNPs on each RAD tag segregates in the type of *ab*×*aa* and the other in *aa*×*ab*, where the first two letters denote the SNP genotype of the female *P. deltoides*, and the last two indicate the genotype of the male *P. simonii* (Figure 1A). These RAD tags allow for performing a two-point linkage analysis of the female or male species or even both (Figure 1B,C). First, we used the SNPs of *ab*×*aa* to construct the female linkage map with the JoinMap 4.1 software [29]. Apparently, not only the RAD tags but also the SNPs of *aa*×*ab* were classified into the same linkage groups with the same orders as the SNPs of *ab*×*aa* in the female map. Second, the two genotype datasets of *ab*×*aa* and *aa*×*ab* were merged into one dataset for constructing an integrated map of the two parents after a transformation procedure. In each female linkage group map, when two adjacent SNPs of *ab*×*aa* were in the repulsion phase, the progeny genotype of “ab” at the second SNP was transformed into “aa” and the genotype of “aa” into “ab” (Figure 2). It is easily verified that the same female recombination fraction for the two SNPs can be calculated from the transformed data [16]. Similarly, data transformation along the order of RAD tags was also performed for the SNPs of *aa*×*ab* in the same linkage group. We can see that each of the two transformed datasets is like those from the traditional BC population, and so is the merged dataset. The merged dataset for each linkage group was used to construct the integrated group map with the MapMaker software [5]. Third, the linkage maps of RAD tags were plotted in a WMF format using Kosambi’s mapping function [30] and FsLinkageMap. Finally, the maps were further changed into a PDF format using the tool Mayura Draw (http://www.mayura.com, accessed on 5 July 2022).

After the above procedures, the female linkage map and an integrated linkage map were generated. Meanwhile, the male linkage map was obtained by just using the SNPs of *aa*×*ab* with JoinMap. In order to investigate the performance of the three linkage maps, we compared their collinearities with a reference sequence. Additionally, for a linkage group, the linear orders of the three linkage maps were compared by calculating the likelihoods with the transformed and merged genotype data using MapMaker.

## 3. Results

### 3.1. Selected RAD Tags

After filtering the RAD-seq data, an average of 4.61 Gb HQ data were obtained to call the SNP genotypes for each individual. A total of 48,938 SNPs were detected and genotyped in the F_1_ hybrid population by aligning the HQ reads to the reference genome of *P. simonii*. Like the results in our previous studies [1,20], most SNPs were segregated in the types of *ab*×*aa* and *aa*×*ab*, amounting to 29,466 and 18,863, respectively. From the two types of 1:1 segregated SNPs, a total of 1154 RAD tags were obtained for constructing an integrated linkage between the two parents. Each RAD tag had two SNPs within a range of <1 Kb, one of which was segregated in the type of *ab*×*aa* and the other in *aa*×*ab* (Figure 1A). Moreover, the distance between two adjacent RAD tags was required to be ≥100 Kb so that the crossover events between them can be frequently identified.

### 3.2. Integrated Linkage Map of P. deltoides and P. simonii

First, the female parental linkage map of *P. deltoides* was built with the SNP dataset of *ab*×*aa*, which contained 19 linkage groups under the logarithm of odds (LOD) thresholds ranging from 5 to 17 and thus perfectly matched the karyotype of *Populus* (Appendix A). Correspondingly, the RAD tags were also classified into 19 linkage groups. For each linkage group of RAD tags, the SNP genotype datasets of *ab*×*aa* and *aa*×*ab* were transformed and merged to construct the integrated group map, where the RAD tags were denoted as “RT” plus the serial number along the reference genome of *P. simonii* (Figure 3). As a result, the total length of the integrated linkage map was 5255.49 cM, with an average genetic distance of 4.63 cM between the adjacent RAD tags and a linkage group length ranging from 164.42 cM for chromosome 19 to 694.79 for chromosome 1 (Table 1).

Besides the construction of the female parent linkage map, we also used the SNP dataset of *aa*×*ab* to construct the male parent linkage map (Appendix A). For the two parental linkage maps and the integrated linkage map, the number of markers and the length of each LG are presented in Table 1. The numbers of SNPs or RAD tags within the same LG were equal for the three linkage maps. Except for LGs 8 and 12, most female LG lengths were consistently greater than those of the male species. Moreover, the total length of the female map reached 5607.34 cM, much greater than that of the male (4752.21 cM), indicating that the recombinant events were more frequent in the female *P. deltoides* than in the male *P. simonii*. From the integrated map, it is easily found that each LG length was between those in the female and male linkage maps.

### 3.3. Advantage of the Integrated Map over the Two Parental Maps

In order to investigate the advantage of the integrated linkage map over the two parental linkage maps, we first compared their consensuses with the genomic sequences of *P. simonii* [25] and *P. deltoides* [31]. We found that a total of 68 local regions on the integrated linkage map were discordant with the *P. simonii* genomic sequence, which contained ~15% (173) of all the RAD tags (Appendix A; Table 2). However, there were more discordant regions (DRs) on each parent linkage map, with 119 and 88 DRs containing ~31% (361) and ~21% (239) markers on the female and male maps, respectively. Moreover, the number of DRs within LGs in the integrated map was largely less than or equal to the numbers in both the female map and the male map, except for LGs 3, 4, 15, and 16. Meanwhile, the numbers of markers in the DRs within the LGs basically had the same situation.

To compare the consensuses with the genome of *P. deltoides* “I-69” among the three linkage maps, the RAD tags were mapped to this genomic sequence. Consequently, a total of 1134 RAD tags were successfully aligned to the corresponding chromosomes, except for 20 markers mapped to different chromosomes or scaffolds (Appendix A). The DRs and the internal markers in the integrated, female, and male linkage maps were counted for each LG and are listed in Table 3. Similarly, we found that the number of DRs in most integrated LGs was less than or equal to the corresponding numbers in each of the two parental maps, except for LGs 3, 4, and 15. Additionally, the number of markers contained in the DRs showed the same feature as that of the DRs. In total, there were 77 DRs with 197 markers in the integrated map, while there were more DRs with more markers in the female and male maps (Table 3).

In addition to the comparison of consensuses with genomic sequences, we also investigated the statistical characteristics of the marker orders within LGs in the three linkage maps. The likelihood of the marker order in each LG was calculated based on the transformed and merged data of *ab*×*aa* and *aa*×*ab* with the MapMaker software. The result showed that the likelihood of each LG in the integrated map was greater than that in both the female and male maps (Table 4), indicating that, from a statistical point of view, the marker orders are more accurate in the integrated map than in the two parent maps.

## 4. Discussion

Throughout time, genetic statisticians have tried to use fully informative markers as bridges for constructing an integrated map in a full-sib family in outbred species, especially in forest trees [16,17,18]. Here, the fully informative markers include those that segregate into the types of *ab*×*ab* and *ab*×*cd* in a mapping population. However, only a small number of fully informative SNPs were identified in the F_1_ hybrid population of *P. deltoides* and *P. simonii* in our previous linkage mapping studies, so they were not sufficient to build an integrated linkage map but led to two parent-specific linkage maps [1,20,24]. In the current study, we successfully obtained an integrated genetic linkage map of the two parents in the same population, using a novel strategy of selecting specific RAD tags from a large number of SNPs across the population. Besides the selection of the RAD tags, we proposed a method to transform and merge the two genotype datasets of *ab*×*aa* and *aa*×*ab* into one dataset, just like in a traditional BC population, allowing for the use of the famous software MapMaker for linkage mapping. The integrated linkage map not only has some advantages over the two parental linkage maps but also possesses superiority in comparative genomics and QTL mapping.

Compared with the two parent-specific linkage maps, the integrated linkage map was more consensus with the genomic sequences of *P. simonii* [25] and *P. deltoides* [31]. Although the available genomic sequences have some shortcomings due to the complexity of genomic assembling, such consensus reflects a certain degree of authenticity in the order of contigs because the genomic assemblies and genetic maps were obtained using independent technologies. In the process of genomic assembling, the long reads of the two parents obtained with third-generation sequencing technologies were first assembled into contigs. Then, these contigs were further anchored into chromosome-level sequences with the chromosome conformation capture (Hi-C) technology. However, the Hi-C technology is not entirely perfect, as it could produce some errors in a chromosome-level assembly such as artificial inversions and misjoined contigs [32,33]. On the other hand, the order of markers on linkage maps may not be absolutely correct. The reason is that ordering a large number of markers in a linkage group is a difficult, scientific problem, which could produce erroneous orders in local regions [1]. Nevertheless, the consensus between the orders of contigs and genetic markers could validate each other to make sure they are as correct as possible.

In theory, the integrated linkage map provides a powerful genetic resource for identifying QTLs, with additive and dominant effects in the F_1_ hybrid population of *P. deltoides* and *P. simonii*. However, there are no software packages available to directly implement the analysis of QTL mapping. Therefore, new algorithms should be developed to calculate the genetic parameters for a specific position on the integrated map, similar to the traditional QTL mapping methods such as interval mapping (IM) [2] and composite interval mapping (CIM) [34]. Unlike in the F_2_ population from inbred lines, the two haplotypes of an individual can be discriminated from its genotype with the RAD tags. For this reason, the probability of an assumed QTL genotype conditional on the flanking RAD tags could be easily derived, and thus the likelihood of the QTL effects could be established. After that, the calculation for the parameters can be implemented through the expectation–maximization (EM) algorithm [35]. We expect to finish such a statistical analysis in a subsequent study and to detect QTLs with dominant effects for important growth traits in related *Populus* species.

## 5. Conclusions

In this study, a novel strategy was proposed and successfully applied to construct an integrated linkage map for *P. deltoides* and *P. simonii* with next-generation sequencing data. However, the conventional linkage mapping methods for such a hybrid population led to two parent-specific linkage maps, each lacking the full recombinant information of both parents. The integrated linkage map proved to be more accurate than the two parent-specific linkage maps, providing a powerful genetic resource for identifying quantitative trait loci (QTLs) with dominant effects, assembling genomic sequences, and performing comparative genomics in *Populus*. Most importantly, this novel strategy can be used in other outbred species for constructing an integrated genetic linkage map.

## Figures and Tables

**Figure 1 genes-13-01731-f001:**
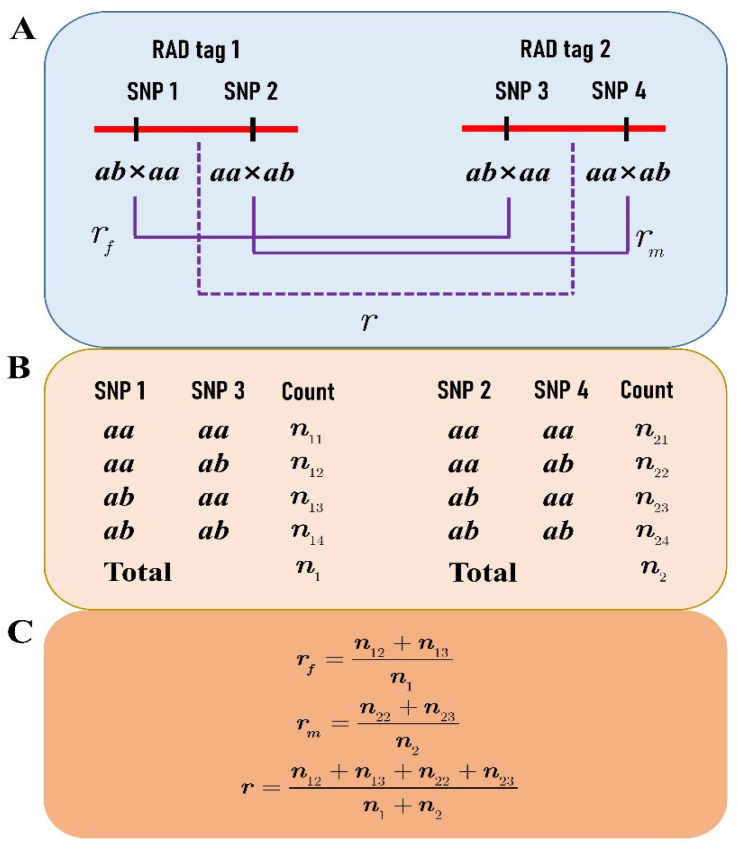
Recombination fraction between two RAD tags: (**A**) each RAD tag contains two SNPs, one segregating in the type of *ab*×*aa* and the other in *aa*×*ab*; (**B**) the genotype counts at SNPs 1 and 3 in the progeny are used to estimate the female recombination fraction, while the counts at SNPs 2 and 4 are for estimating the male recombination fraction; (**C**) under the assumption of the SNP linkage phases in coupling, the formulas are given for calculating the recombination fractions of the female (*r_f_*), the male (*r_m_*), and the combination of both (*r*).

**Figure 2 genes-13-01731-f002:**
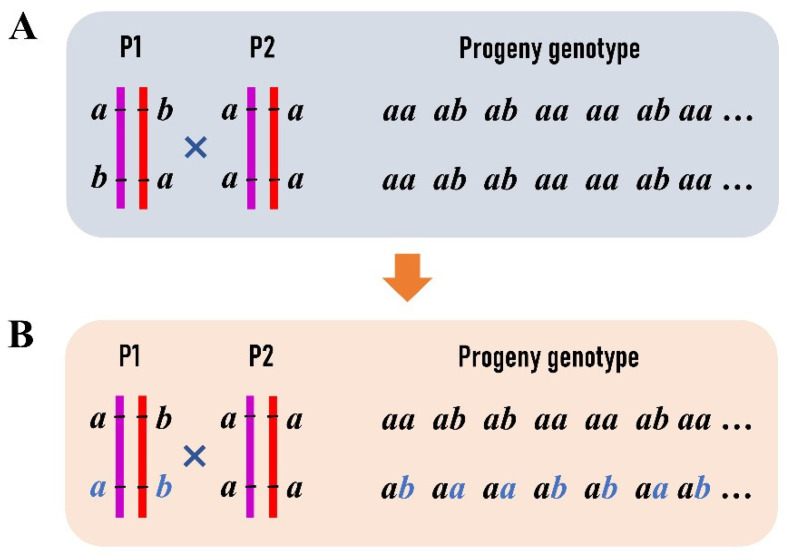
Transformation of progeny genotype data at two linked SNPs in an F_1_ outbred population: (**A**) the linkage phase of the maternal parent is repulsion with haplotypes as ab/ba, where the same letters at different sites represent different alleles; (**B**) at the second SNP site in the maternal parent, allele “a” is transformed into “b” and “b” into “a”. Correspondingly, the genotype of “aa” is transformed into “ab” and “ab” into “aa” in the progeny.

**Figure 3 genes-13-01731-f003:**
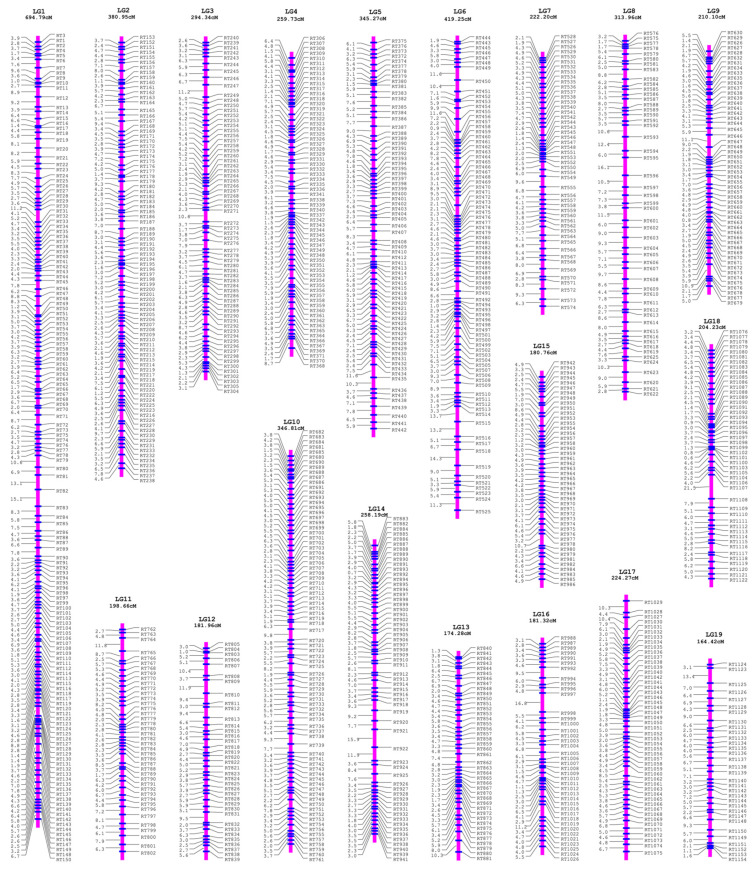
The integrated genetic linkage map of the female *P. deltoides* and the male *P. simonii*. The map included 19 linkage groups from LG1 to LG19 with the LG length presented in cM.

**Table 1 genes-13-01731-t001:** The number of SNPs and the length of linkage groups in the female, male, and integrated linkage maps.

LG	SNP Number	Integrated (cM)	Female (cM)	Male (cM)
1	150	694.79	765.09	585.60
2	88	380.95	392.38	347.07
3	67	294.34	294.25	288.02
4	66	259.73	273.77	231.81
5	71	345.27	403.63	276.24
6	83	419.25	466.13	363.22
7	49	222.20	248.56	187.51
8	51	313.96	298.29	365.90
9	54	210.10	213.75	195.74
10	82	346.81	386.21	294.40
11	41	198.66	233.42	157.57
12	37	181.96	179.05	180.73
13	42	174.28	188.58	156.21
14	60	258.19	257.45	247.65
15	45	180.76	190.43	172.68
16	40	181.32	194.77	166.06
17	49	224.27	249.69	191.77
18	47	204.23	204.40	191.56
19	32	164.42	167.49	152.47
Total	1154	5255.49	5607.34	4752.21

**Table 2 genes-13-01731-t002:** The number of discordant regions with marker number in brackets in each linkage group of the two parent linkage maps and the integrated linkage map. The marker order of the region is discordant with the genomic sequence of *P. simonii*.

LG	Integrated	Female	Male
1	6 (12)	14 (60)	13 (27)
2	7 (17)	11 (31)	7 (22)
3	3 (6)	9 (21)	2 (8)
4	8 (17)	7 (17)	7 (26)
5	5 (16)	10 (28)	6 (18)
6	4 (11)	8 (30)	7 (14)
7	3 (10)	4 (11)	3 (10)
8	3 (8)	4 (16)	6 (13)
9	3 (8)	5 (17)	4 (10)
10	4 (15)	6 (21)	7 (22)
11	1 (2)	5 (15)	2 (4)
12	2 (5)	2 (6)	4 (8)
13	1 (2)	3 (6)	1 (5)
14	3 (6)	6 (17)	4 (8)
15	4 (8)	5 (10)	3 (7)
16	4 (8)	6 (20)	2 (5)
17	3 (13)	5 (14)	4 (16)
18	2 (5)	5 (13)	3 (10)
19	2 (4)	4 (8)	3 (6)
Total	68 (173)	119 (361)	88 (239)

**Table 3 genes-13-01731-t003:** The number of discordant regions with marker numbers in brackets in each linkage group of the two parent linkage maps and the integrated linkage map. The marker order of the region is discordant with the genomic sequence of *P. deltoides* “I-69”.

LG	Integrated	Female	Male
1	8 (16)	17 (50)	15 (33)
2	8 (19)	12 (31)	9 (23)
3	4 (8)	10 (23)	3 (10)
4	6 (16)	5 (10)	8 (24)
5	5 (13)	11 (25)	5 (17)
6	3 (6)	6 (30)	7 (14)
7	4 (12)	4 (15)	4 (12)
8	4 (12)	5 (18)	7 (16)
9	4 (10)	5 (20)	4 (10)
10	5 (13)	6 (20)	9 (20)
11	2 (4)	5 (13)	2 (4)
12	2 (7)	3 (6)	4 (8)
13	3 (6)	4 (8)	3 (9)
14	3 (8)	6 (15)	4 (11)
15	6 (15)	7 (14)	4 (12)
16	3 (11)	6 (19)	3 (6)
17	3 (12)	5 (17)	4 (16)
18	2 (5)	6 (15)	4 (9)
19	2 (4)	3 (6)	3 (6)
Total	77 (197)	126 (355)	102 (260)

**Table 4 genes-13-01731-t004:** The likelihood of maker order within linkage groups of the integrated, female, and male linkage maps, calculated with the transformed and merged dataset of *ab*×*aa* and *aa*×*ab*.

LG	Integrated	Female	Male
1	−6024.86	−6334.99	−6090.84
2	−3471.39	−3570.42	−3515.76
3	−2723.86	−2818.21	−2731.89
4	−2495.26	−2528.05	−3131.32
5	−3041.48	−3147.71	−3061.93
6	−3518.74	−3744.25	−3557.78
7	−2040.54	−2063.68	−2047.55
8	−2573.69	−2648.03	−2700.16
9	−2003.78	−2032.57	−2060.53
10	−3234.3	−3322.26	−3266.26
11	−1787.37	−1849.56	−1816.54
12	−1661.27	−1669.3	−1693.25
13	−1688.35	−1721.56	−1704.13
14	−2345.77	−2410.72	−2383.61
15	−1785.14	−1830.11	−1810.19
16	−1690.17	−1733.3	−1699.84
17	−2083.31	−2112.19	−2141.12
18	−1878.04	−1920.67	−1895.44
19	−1473.79	−1492.22	−1509.77

## Data Availability

The RADseq data for all individuals are available in the NCBI SRA database (http://www.ncbi.nlm.nih.gov/Traces/sra, accessed on 1 May 2022) with accession numbers listed in Appendix A.

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
