# Peer review of "A Novel Strategy for Constructing an Integrated Linkage Map in an F1 Hybrid Population of Populus deltoides and Populus simonii"

_genes, 2022, doi:10.3390/genes13101731_

Round 1
Reviewer 1 Report
Manuscript title: A Novel Strategy for Constructing an Integrated Linkage Map in an F1 Hybrid Population of Populus deltoides and Populus simonii
Manuscript ID: genes-1898865
Journal: Genes
The aim of this study was to propose a novel strategy to construct an integrated linkage map that contained both parental recombinant information between adjacent markers in an F1 hybrid population of Populus deltoides and Populus simonii. The idea is sound and the manuscript is well written. Abstract is informative. Material and methods contain details which help other researchers to follow easily. Results and discussion are correlated. Academic English language is fine. However, some suggestions need to be considered:
The aim should be written clearly at the end of introduction section;
Statistical analysis should be performed;
Discussion is weak and must be improved;
Used references are OLD and should be updated.
Author Response
Response to Reviewer 1
The aim of this study was to propose a novel strategy to construct an integrated linkage map that contained both parental recombinant information between adjacent markers in an F1 hybrid population of Populus deltoides and Populus simonii. The idea is sound and the manuscript is well written. Abstract is informative. Material and methods contain details which help other researchers to follow easily. Results and discussion are correlated. Academic English language is fine. However, some suggestions need to be considered:
RE: Thank you very much for your constructive and positive comments.
The aim should be written clearly at the end of introduction section;
RE: Thank you very much for this suggestion. As you indicated above, the aim of this study was to propose a novel strategy to construct an integrated linkage map that contained both parental recombinant information. We carefully checked and found that we expressed this point clearly in lines 66-69. In order to emphasize this point completely, we added a sentence at the end of introduction section as follows: “Furthermore, the proposed strategy could be applied in other outbred species for constructing an integrated linkage map with an F1 hybrid population”.
Statistical analysis should be performed;
RE: Thank you very much for this suggestion. We have presented the formulas for computing the recombination fractions of the female, the male, and the combination of both. As for the other analyses such as linkage grouping and marker ordering, they are very complicated and incorporated in the mapping software tools such as JoinMap and MapMaker. These methods are well described in the traditional references for linkage mapping analysis and can be found in the manuals of the two software.
Discussion is weak and must be improved;
RE: Thank you very much for this suggestion. In the first paragraph, we discussed that our proposed mapping strategy is novel and we implemented it in our hybrid population. In the second paragraph, we discussed the integrated linkage map could be used to correct genome assembly. In the third paragraph, we discussed the integrated map was a powerful resource for QTL mapping but need develop specific models in the further. These discussions included the novelty of our strategy and the important applications of the integrated linkage map.
Used references are OLD and should be updated.
RE: Thank you very much for this suggestion. We carefully checked and found that the references of [2] (1989), [5] (1987), [12] (1994), [16] (1997), [30] (1943), [34] (1994), and [35] (1977) are possibly old, but they belonged to the pioneer works in the fields of genetic linkage analysis and QTL mapping. Therefore, we cited these references as many other authors did in the fields.

Reviewer 2 Report
Dear authors,
thank you for a presented draft.
From my side I have just few comments:
row 161 Populus did you mean the entire genus Populus or just a specific related species? The same in row 277.
Fig.3 Maybe is sufficient present just a part of the picture (lenght is summarized in Tabel 1. so could be enough and much easier to read).
Thats all my comments.
I wish you good luck with your future research!
Author Response
Response to Reviewer 2
Dear authors,
thank you for a presented draft.
From my side I have just few comments:
row 161 Populus did you mean the entire genus Populus or just a specific related species? The same in row 277.
Fig.3 Maybe is sufficient present just a part of the picture (lenght is summarized in Tabel 1. so could be enough and much easier to read).
Thats all my comments.
I wish you good luck with your future research!
RE: Thank you very much for your positive comments. 1) For clarifying, we deleted “in Populus” in line 161 and changed “Populus” into “related Populus species” in line 277. 2) The length of each linkage group presented in Figure 3 was just like what we did in our previous linkage mapping works. 3) Table 1 was used to compare the lengths of linkage groups of the three maps.